# Quantum Variational vs. Quantum Kernel Machine Learning Models for Partial Discharge Classification in Dielectric Oils

**DOI:** 10.3390/s25041277

**Published:** 2025-02-19

**Authors:** José Miguel Monzón-Verona, Santiago García-Alonso, Francisco Jorge Santana-Martín

**Affiliations:** 1Electrical Engineering Department (DIE), University of Las Palmas de Gran Canaria, 35017 Las Palmas de Gran Canaria, Spain; francisco.santana@ulpgc.es; 2Institute for Applied Microelectronics, University of Las Palmas de Gran Canaria, 35017 Las Palmas de Gran Canaria, Spain; santiago.garciaalonso@ulpgc.es; 3Department of Electronic Engineering and Automatics (DIEA), University of Las Palmas de Gran Canaria, 35017 Las Palmas de Gran Canaria, Spain

**Keywords:** partial discharges, mineral oils, quantum machine learning, quantum variational model, quantum kernel model, image processing with AI

## Abstract

In this paper, electrical discharge images are classified using AI with quantum machine learning techniques. These discharges were originated in dielectric mineral oils and were detected by a high-resolution optical sensor. The captured images were processed in a Scikit-image environment to obtain a reduced number of features or qubits for later training of quantum circuits. Two quantum binary classification models were developed and compared in the Qiskit environment for four discharge binary combinations. The first was a quantum variational model (QVM), and the second was a conventional support vector machine (SVM) with a quantum kernel model (QKM). The execution of these two models was realized on three fault-tolerant physical quantum IBM computers. The novelty of this article lies in its application to a real problem, unlike other studies that focus on simulated or theoretical data sets. In addition, a study is carried out on the impact of the number of qubits in QKM, and it is shown that increasing the number of qubits in this model significantly improves the accuracy in the classification of the four binary combinations studied. In the QVM, with two qubits, an accuracy of 92% was observed in the first discharge combination in the three quantum computers used, with a margin of error of 1% compared to the simulation obtained on classical computers.

## 1. Introduction

Partial discharge (PD) detection in transformer dielectric oils is justified by the need to reduce costly transformer breakdowns, extend their lifespan, optimize preventive maintenance, and eliminate network failures, all of which can have significant economic impacts. A steady and sustained rise in the number of publications on PD source classification using machine learning algorithms can be seen in the period from 2010 to 2023 [1].

PD occurs when high voltage is applied to materials in any state, whether solid, liquid, or gaseous. It is a complex physical process that exhibits randomly distributed properties and produces phenomena such as light, sound, and high-frequency electromagnetic waves, releasing electrical charges [2].

In situ experimental images of transformer oil spaces are extremely complex. In this work, the tests have been performed in the laboratory with oil samples extracted from the transformer. Consider that the main objective of this work is to explore the feasibility and potential of quantum machine learning models for the classification of electrical discharges in dielectric oils, using a controlled laboratory environment.

Although our current work is not focused on in situ studies, it could lay the groundwork for future research in that direction. This laboratory setup offers the advantage of immunity to electromagnetic interference.

The most widely used AI-based machine learning algorithms currently used to identify PDs in electrical transformers are derived using support vector machines (SVMs) [3], followed by artificial neural networks (ANNs) [4] and convolutional neural networks (CNNs) [5]. All these methods use classical computing. In ref. [6], it is indicated that before machine learning algorithms are tested on specific problems, there are no inherent or predefined differences that allow us to affirm that one machine learning algorithm is better than another. Following the current trend in the use of SVM techniques for the analysis and search for patterns in difficult-to-classify environments, the so-called kernel trick is used; through it, an attempt is made to find a series of hyperplanes where it is easier to find certain values. Once these SVM techniques are known, the aim is to transfer this knowledge to quantum computing.

Below, we review the current state of quantum kernel models (QKMs), quantum variational models (QVMs), the use of currently employed fault-tolerant quantum computers, and their potential theoretical and experimental advantages as well as their limitations.

The QKM is an area of AI in which the advantage of quantum computing has been explored. According to ref. [7], quantum kernels can be used for supervised learning, showing that a quantum computer can classify data in a high-dimensional feature space more efficiently than classical methods.

In ref. [8], it is explained how QKMs can capture complex relationships and patterns in data that classical kernels might not be able to identify. In this way, an SVM using a quantum kernel can better classify new data and make more accurate predictions. This enables hybrid computing, where a quantum computer implements a quantum kernel that is then run on a classical computer.

In ref. [9], it is noted that as the problem size increases, the differences between kernel values become smaller and smaller, and more measurements are required to distinguish between the elements of the kernel matrix.

In ref. [10], the number of evaluations when solving the dual problem is quantified in a number of quantum circuit evaluations with an order of magnitude given by Equation (1), where M represents the size of the data set and ϵ is the accuracy of the solution compared to the ideal result, which can only be obtained theoretically with exact values. That is, the time required to solve the dual problem using quantum circuits increases polynomially with the size of the data set *M* and is inversely proportional to the square of the accuracy ϵ.

The dependence on *M* poses a major challenge for problems with large data sets. In ref. [10], an improvement with the primal problem with the kernel is shown using a generalization of a classical algorithm known as Pegasos, resulting in a smaller number of evaluations which, using Landau notation, is shown in Equations (1) and (2).(1)OM4.67∈2(2)OminM2∈6,1∈10

In ref. [11], it is explained that the QKM approach is more natural and suitable for quantum theory compared to the attempt to adapt quantum theory to fit the structure of classical neural networks, which is a more popular but less natural approach.

Thus, instead of optimizing classical parameters with QVM, which presents certain complex problems such as the choice of ansatz and the appearance and treatment of sterile plateaus, the QKM approach avoids these problems, although it requires calculating pairwise distances between data points, which implies a high computational cost.

As shown by the results reported in ref. [12], quantum algorithms can outperform classical algorithms in optimization problems, which are central to supervised learning. A review of several quantum optimization algorithms is also conducted in ref. [12], and their potential to outperform classical methods in machine learning tasks is pointed out, highlighting certain practical applications and preliminary experiments.

In ref. [13], it is shown how a quantum perceptron can be simulated on a quantum computer, suggesting that QVMs could be trained and run more efficiently than their classical counterparts in certain cases. In ref. [14], it is suggested that a QVM with the ability to process classical and quantum data, trainable through supervised learning, could be run on an intermediate-scale quantum computer.

It is interesting to note that in ref. [15], it is demonstrated that quantum computers can handle and process structured data more efficiently in some specific cases. The quantum algorithm can, in theory, outperform classical methods in performing principal component analysis (PCA) on large data sets.

Quantum computing is a technique based on random phenomena that occur at the atomic scale. This computing uses the properties of quantum mechanics such as quantum superposition and entanglement. Its basic unit of information is the qubit, similar to the bit in classical computing.

It is important to highlight the greater computational power of quantum computing because qubits can exist simultaneously in multiple states. In quantum computing, the computational power increases exponentially as the number of qubits increases, which can be compared to classical computing where this increase is linear as the number of bits increases.

The execution speed in quantum computing is greater than in classical computing due to the principles on which it is based, such as quantum superposition and entanglement, which give rise to parallel computing.

Therefore, the execution of these algorithms can be repeatedly invoked many times, obtaining an acceptable probabilistic response in certain practical problems.

Among these algorithms, the best known is Shor’s algorithm [16], which is a reference used in the factorization of prime numbers with a polynomial complexity $O\left({\left(logN\right)}^{3}\right)$ compared to the generalized prime number algorithm with a complexity OexplogN13loglogN13 [17].

Another algorithm of great interest in quantum computing is Grover’s algorithm [18], which has been shown to be fundamentally useful in searching for a given element in an unstructured database with a theoretical complexity of $O\sqrt{N}$, compared to $O\left(N\right)$. The Long–Grover algorithm [19] is a variant of Grover’s quantum search algorithm that is able to handle situations where the exact number of solutions in the unstructured database is not known. This new algorithm maintains the same theoretical efficiency $O\sqrt{N}$ for database dimension *N* as Grover’s algorithm, but with a better ability to tolerate uncertainty in the proportion of solutions, making it more robust and practical for certain types of search problems.

In the analysis undertaken in ref. [20], the question is raised as to whether and how quantum computing can actually boost machine learning using real-world classical data sets. The main technical limitations and challenges associated with noisy intermediate-scale quantum (NISQ) computers are then addressed.

An analysis of the quantum computing landscape is discussed in ref. [21], where it is stated that current quantum computers are not perfect due to decoherence in qubits caused by environmental noise, but they can perform certain calculations or solve problems that are beyond the reach of the best classical computers available today. It is argued that the main limitation will be the ability to maintain precision in quantum operations as circuits become larger and more complex. In ref. [22], it is explained how to experimentally extend the coherence time of logical qubits by almost an order of magnitude.

Another challenge that exists today is to maintain a large number of entangled qubits in a stable manner. Quantum computers that address this challenge include the IBM Kyoto [23], IBM Brisbane [24,25], and Google’s Sycamore quantum computer [26], among others.

To our knowledge, the use of QKM and QVM techniques has not been applied in the analysis of real cases in PD image detection in transformer oils using optical sensors. In this paper, a comprehensive study of PDs originating from bubbles present in dielectric mineral oil is carried out. These discharges are precursors of the arc breakdown and therefore represent a method for diagnosing the state of mineral oil before such a breakdown occurs.

In this paper, images captured in a high-voltage laboratory are processed by selecting a number of significant features. For this purpose, the Scikit-image environment [27] is used. Two quantum classifier models, QKM and QVM, are developed. These models are implemented in the Qiskit development environment [28].

In this article, we focus on the use of quantum computing to address the problem of image classification in transformer dielectric oils. The images used in this article are classified into four categories: images with partial discharges, images without partial discharges, images with electric arc breaking, and images with gas bubbles after arc breaking.

The effectiveness of the trained QKM and QVM classifiers is evaluated with images not used during the training process.

These models were run on three fault-tolerant physical quantum computers, each with 127-qubit superconducting processors: IBM Osaka, IBM Brisbane, and IBM Kyoto. The measurements obtained using quantum computers were then compared with the results of simulations obtained using classical computing.

The main contribution of this paper is that for the first time, two quantum machine learning models, QVM and QKM, are applied and compared for the classification of electrical discharge images in dielectric oils, using real data obtained with a high-resolution optical sensor.

The novelty of this work can be summarized in the following points: The work is applied to a real problem, unlike other previous studies that focus on simulated or theoretical data sets. In addition, a study is carried out on the impact of the number of qubits in QKM, and it is shown that increasing the number of qubits in this model significantly improves the accuracy in the classification of the four binary combinations of the classes. On the other hand, real quantum computers are used, and the models are implemented and executed on three fault-tolerant IBM quantum computers, demonstrating their operation on real quantum hardware and providing results comparable to classical simulations. This work also provides transparency and reproducibility by creating a repository on Zenodo [29], with a detailed README, where the images of the electrical discharges used, the Jupyter Notebooks 7.0.8 for the extraction of the features, and the Jupiter Notebooks with the Python 3.12.4 programming of QVM and QKM with the respective figures have been published, so that the scientific community can access and use them.

This article is divided into the following sections: Section 2, Image Processing and Feature Extraction Method; Section 3, Quantum Machine Learning with Variational Circuits (Quantum Variational Model, QVM); Section 4, Support Vector Machine, SVM; Section 5, Quantum Kernel Model, QKM; Section 6, Overall Flowchart. Finally, in Section 7, Conclusions, the main conclusions are presented.

## 2. Image Processing and Feature Extraction Method

In this work, an analysis of the PDs originated from bubbles present in dielectric mineral oils is performed. For this purpose, a high-resolution image sensor is used. The PDs detected with this sensor were validated using a standard electrical detection system using a discharge capacitor, according to the IEC60270 standard [30]. All images used in this paper were previously obtained by the authors [31]. These images were used to characterize and train the quantum circuits in Section 3 and Section 5.

From the extraction of features in machine learning, relevant values are obtained from the obtained experimental images, speeding up the computing process without losing information. This reduces the required memory and computing time and improves the accuracy of the model.

In ref. [1], the main techniques used to date are summarized. Those based on statistical characteristics are highlighted, as well as the technique based on principal component analysis (PCA) due to its capacity to reduce dimensionality and identify key variables, among others. This is crucial when working with a limited number of qubits that correspond to the current limitations of quantum technology. The method used in this article is explained below.

Figure 1a presents four images, each corresponding to one of the four classes used: class 0 for partial discharge (PD), class 1 for no discharge (NOPD), class 2 for electric arc breaking (ARC), and class 3 for gas bubbles after arc breaking (BREAK). Figure 1b shows the experimental image collection device. From these images, features are extracted that reduce the number of qubits needed to perform quantum analyses.

Features are basic properties that characterize and simplify experimental images of electrical discharges. The goal is to work with as few qubits as possible. For this reason, the extraction of features from images has been reduced to a maximum of thirteen.

To process the images captured in the high-voltage laboratory and select their features, the Scikit-image environment is used. An explanation of the entire Scikit-learn environment, which includes working with images in multiple formats and provides tools for transforming, analyzing, and improving images, is provided in ref. [27]. This includes filtering functions, geometric transformations, edge detection, segmentation, and color manipulation, among others.

To analyze the characteristic features of electrical discharges in images captured with a high-quality camera, a region of interest (ROI) selection and analysis process was first performed. Thus, the ROI was defined as a square centered on the image with a side of 100 pixels. To do this, the coordinates of the center of the image were calculated, and the vertices of the square were located (Figure 2a). The ROI within this square was converted to greyscale to facilitate the analysis. The ROI is shown in a red frame.

Then, the mean and standard deviation of the pixel intensities within the ROI were calculated. Using these values, a threshold was set as the mean plus two times the standard deviation. Pixels whose intensity exceeded this threshold were identified and their coordinates determined. The mean of these coordinates was then calculated to obtain the centroid of the high-intensity region.

To highlight pixels exceeding the threshold, the original image was modified by highlighting these pixels in red. In addition, the area of the highlighted region was calculated in terms of the number of pixels, and the centroid of this region was determined. Finally, the coordinates of the centroid were adjusted with respect to the originally selected ROI (see Figure 2a).

Key features obtained in this image analysis include the area in pixels of the highlighted region, the centroid coordinates in both the ROI and the original image, and the intensity statistics of the ROI. All these features are normalized to the interval [0, 2*π*].

The thirteen features of the images and the class to which they belong are as follows: the area in pixels, the centroid coordinates (centroid_x, centroid_y), the adjusted centroid coordinates in the ROI (centroid_x_roi, centroid_y_roi), the means of the coordinates (mean_coords_x, mean_coords_y), the size of the side of the square (side_px), the dimensions of the image (image_width, image_height), the mean intensity (mean_intensity), the standard deviation of the intensity (std_intensity), the threshold (threshold), and finally the class to which it belongs.

Figure 2b and Figure 3b show the preprocessing results with the acquisition of the features for the PD and BREAK classes, respectively.

From the images of each binary combination between classes, a graphic study of the relationship between the possible pairs of features of the images, which turn out to be 13 × 13 graphs, was performed. This allows a preliminary study of the relationship between the different features of each class. This visualization allows for a first analysis, and to identify patterns and differences between the binary classes of images corresponding to each combination of classes.

Figure 4 presents a pairwise plot illustrating the relationships between five of the thirteen normalized image features, highlighting the two classes using colors for the PD_BREAK combination shown in Figure 1a.

Likewise, three other binary combinations were analyzed, PD_NOPD, PD_ARC, and BREAK_NOPD. Of all the possible binary combinations, these three are analyzed in this article because they are the most significant in the study of partial discharges.

Two machine learning methods were used, QVM and QKM. The first is considered in Section 3 and uses a trained variational quantum circuit to distinguish each class for each of these binary combinations. The second is considered in Section 5 and uses SVM by estimating the quantum kernel corresponding to each of these binary combinations.

## 3. Quantum Machine Learning with Variational Circuits (Quantum Variational Model—QVM)

### 3.1. Introduction

The first model used to classify PDs is known as a variational quantum circuit (see the red block in Figure 5). This red block has two clearly differentiated parts.

The first part corresponds to the encoding of the chosen features of the images within the quantum circuit. The features are represented by the variable *x*[*i*], where *i* is the number of chosen features that are normalized real numbers in the interval [0, 2*π*]. They are introduced into the circuit through the logic gate *U* that performs a rotation around the z axis.

The second part within this red block is a series of quantum parameters *theta*[*j*] = *θ*[*j*], where *j* depends on the complexity of the circuit, its number of qubits, and its depth. In simulations and executions in real quantum circuits, we used circuits with 10 and 11 quantum parameters.

These parameters, just like in neural networks, are fitted in the training phase of the network according to a cost function $C\left(\theta_{i}^{p}\right)$ that is attempted to be minimized at each step *p* by means of an optimization algorithm (see blue block in Figure 5). This algorithm updates the parameters $C\left(\theta_{i}^{p+1}\right)$ at the next step *p* + 1 in the quantum circuit. The cost function is built by comparing the measurement phase in the quantum circuit inside the red block with the expected value known in the supervised training.

To solve the optimization problem, the SciPy package [32] was used, which is a library of numerical routines for the Python programming language. Two optimization algorithms were employed: the simultaneous perturbation stochastic approximation (SPSA) algorithm, which uses a stochastic approximation to estimate the gradients through simultaneous perturbations in all dimensions of the parameter space, which reduces the number of necessary evaluations [33], and the constrained optimization by linear approximations (COBYLA) algorithm [32,34], which does not require derivatives of the objective function or the constraints.

The number of iterations in both methods is between 50 and 100. The main challenge when using this method is to find the structure of the quantum circuit called ansatz. For this task, we used the Qiskit library [35], an open-source Python library for developing quantum computing programs that provides tools for building, simulating, and running quantum circuits on IBM quantum computers. This library allows users to work with quantum algorithms and optimize solutions for complex problems which greatly helps in finding the optimal parameters.

Another important challenge encountered is sterile plateaus. These are regions of the parameter space where the gradient is almost zero, making optimization very slow and difficult.

### 3.2. Cost Function

To obtain the cost function represented in the blue block shown in Figure 5, the concept of cross-entropy loss is followed [36]. In binary classification problems, where the output can be 0 or 1, the cross-entropy loss function is used to measure the difference between the true label and the probability predicted by the model. Its formulation is(3)$$L\left(y,\hat{y}\right)=-\left[y\log\left(\hat{y}\right)+\left(1-y\right)\log\left(1-\hat{y}\right)\right]$$
where *y* is the true label (0 or 1), and $\hat{y}$ is the predicted probability for label 1.

This equation penalizes incorrect predictions for both label 0 and label 1. If the true label is *y* = 1, the term $y\log\left(\hat{y}\right)$ dominates and penalizes the model if it predicts a low probability for label 1. On the other hand, if *y* = 0, the term $\left(1-y\right)\log\left(\hat{y}\right)$ dominates and penalizes the model if it predicts a high probability for label 1 when the true label is 0.

In this work, to simplify the implementation in programming, the cost function focuses only on the probability assigned to the correct class. This can be represented as follows:(4)$$L\left(y_{i},{\hat{y}}_{i}\right)=-\log\left({\hat{y}}_{i}\right)=-\log\left(P\left(y_{i}|x_{i},\theta \right)+\in \right)$$
where *y_i_* is the correct label, ${\hat{y}}_{i}$ is the probability predicted by the quantum circuit designed for that label, and ϵ is a small nonzero value added to avoid logarithms of zero. $P\left(y_{i}|x_{i},\theta \right)$ represents the probability that the quantum circuit assigns a label *y_i_*, which can be 0 or 1, to a data *x_i_*, where *θ* represents the variational parameters that control the quantum circuit.(5)$$P\left(y_{i}|x_{i},\theta \right)=\frac{\sum_{bitstring,\;parity(bitstring)=y_{i}}counts(bitstring)}{total\;shots}$$

For example, for a two-qubit bitstring, a particular bitstring belongs to the set of possible outcomes {00, 01, 10, 11}; the parity is given as parity(00) = 0, parity(01) = 1, parity(10) = 1 and parity(11) = 0. If the particular measurement results of the quantum computer were, for example, $\mathrm{results}=\left\{‘00’:2000,‘01’:250,‘10’:250,‘11’:1500\right\}$, then total shots=2000+250+250+1500=4000, so the probability of obtaining parity 0 is *P*(0) = (2000 + 1500)/4000 = 0.87, and the probability of obtaining parity 1 is *P*(1) = (250 + 250)/400 = 0.13.

Finally, the cost function $C\left(\theta \right)$, which averages the cross-entropy loss for all data samples, *x_i_*, is expressed as(6)$$C\left(\theta \right)=\frac{1}{N}\sum_{i=1}^{N}L\left(y_{i},{\hat{y}}_{i}\right)=-\frac{1}{N}\sum_{1=1}^{N}\log\left(P\left(y_{i}|x_{i},\theta \right)+\in \right)$$
where *N* is the number of data points in the training set.

### 3.3. Structures of the Circuits and Quantum Gates Used

The four stages of how the supervised machine learning problem was approached, its adaptation, transformation, and resolution, using quantum computing, are summarized below. The quantum gates used are also detailed, both in the simulations carried out on the classical computer and on the quantum computer.

The first stage involves mapping the classical problem to its quantum computer formulation. In this stage, the problem is translated into a format that can be processed by a quantum computer. To do this, quantum circuits are created that represent the problem to be solved. This process can be complex and often requires specialized tools. In this work, Qiskit version 1.0 [37] was used, an IBM framework for quantum computing that offers application programming interfaces (APIs) that facilitate the creation of these circuits (Figure 6 and Figure 7).

The second stage is known as circuit transpilation. Once the quantum circuit has been created, it needs to be adapted to be executable on specific quantum hardware. This stage involves rewriting or transforming the original circuit into a version that is compatible and optimized for the available hardware, using gates specific to that hardware (see Figure 8, which corresponds to the transpilation of the circuit in Figure 7). Transpilation transforms it so that only the instructions are available on a chosen backend. They are used and optimized to minimize the effects of noise [38].

The third stage concerns the execution and evaluation of the quantum circuits. In this stage, the transpiled quantum circuit was executed in a quantum simulator, using the Qiskit environment in Python, and additionally on three real physical quantum computers (IBM Osaka, IBM Brisbane, and IBM Kyoto). During this phase, the final measurements leading to the necessary quantum calculations were carried out.

The fourth and final stage involves the post-processing of results. The results obtained from the execution of the quantum circuit are processed and analyzed to find a solution to the original problem posed. This allows the interpretation of the quantum results and the conversion to a format compatible with the classical problem.

The relationships between the quantum gates used in the circuits are shown in Figure 6, Figure 7 and Figure 8. Their matrix expressions are presented in Equations (7)–(10).

The Hadamard gate *U_2_* and *H* gate are represented by Equation (7), the generic rotation gate *U* is defined in Equation (8), and the *z*-axis rotation gate, *P = R_Z_*, is described by Equation (9). The control gate *CX* between two qubits is specified in Equation (10).(7)$$U_{2}\left(\emptyset ,\lambda \right)=\frac{1}{\sqrt{2}}\left(\begin{array}{cc} 1 & -e^{i\lambda }\\ e^{i\phi } & e^{i\left(\phi +\lambda \right)} \end{array}\right);\;U_{2}\left(0,\pi \right)=\frac{1}{\sqrt{2}}\left(\begin{array}{cc} 1 & 1\\ 1 & -1 \end{array}\right)=H$$(8)$$U\left(\theta ,\emptyset ,\lambda \right)=\left(\begin{array}{cc} cos\left(\frac{\theta }{2}\right) & -e^{i\lambda }sin\left(\frac{\theta }{2}\right)\\ e^{i\phi }sin\left(\frac{\theta }{2}\right) & e^{i\left(\phi +\lambda \right)}cos\left(\frac{\theta }{2}\right) \end{array}\right)$$(9)$$U\left(0,0,\lambda \right)=\left(\begin{array}{cc} 1 & -e^{i\lambda }\cdot 0\\ 1\cdot 0 & e^{i\lambda }\cdot 1 \end{array}\right)=\left(\begin{array}{cc} 1 & 0\\ 0 & e^{i\lambda } \end{array}\right)=P\left(\lambda \right)=R_{Z}\left(\lambda \right)$$(10)$$CX=\left(\begin{array}{cccc} 1 & 0 & 0 & 0\\ 0 & 1 & 0 & 0\\ 0 & 0 & 0 & 1\\ 0 & 0 & 1 & 0 \end{array}\right)$$

The ECR gate is a two-qubit gate that performs a controlled operation in the context of cross-resonance. It is one of the native gates in IBM quantum computers and is usually defined in terms of more basic operations, taking into account the interaction between two qubits ([38,39]). Its matrix expression is given in Equation (11). Furthermore, the SX (square root of *X*) gate, whose expression is given in Equation (12), when applied twice, is equivalent to the Pauli *X* gate, as shown in Equation (13).(11)$$ECR=\left(\begin{array}{cccc} \frac{1}{\sqrt{2}} & \frac{-1}{\sqrt{2}} & 0 & 0\\ 0 & 0 & \frac{1}{\sqrt{2}} & \frac{-1}{\sqrt{2}}\\ \frac{1}{\sqrt{2}} & \frac{1}{\sqrt{2}} & 0 & 0\\ 0 & 0 & \frac{1}{\sqrt{2}} & \frac{1}{\sqrt{2}} \end{array}\right)$$(12)$$SX=\frac{1}{2}\left(\begin{array}{cc} 1+i & 1-i\\ 1-i & 1+i \end{array}\right)$$(13)$$X=\left(\begin{array}{cc} 0 & 1\\ 1 & 0 \end{array}\right)$$

### 3.4. Quantum Variational Model (QVM) Optimization

This section details the training process of the two types of quantum circuits used. The first type of circuit, which uses 10 parameters (see Figure 6), was used to perform the binary classification between the combined classes (PD and NOPD). The second type of circuit, with 11 parameters (see Figure 7), was used for the binary classification in the following three combined classes (PD and BREAK), (PD and ARC), and (BREAK and NOPD).

The final optimal parameters for each of these binary combinations, after the optimization process, are presented in Table 1, corresponding to the combinations PD_NOPD, PD_BREAK, PD_ARC, and BREAK_NOPD, respectively.

Since the execution time on the quantum computer is currently limited on the available IBM, it was decided to perform this optimization step to determine the parameters in a simulation using a classical computer, solving the corresponding quantum circuits. The optimal accuracy values for both the training and test sets are shown in the last row of Table 1. The maximum accuracy values on the test set were 95% for PD_BREAK combination, 93% for PD_NOPD combination, and higher than 82% for PD_ARC and BREAK_NOPD combination.

As previously described, Figure 5 shows the blue block where the cost function is calculated at each iteration. The evaluation of the cost function $C\left(\theta_{i}^{p}\right)$ is performed by passing all the images, together with their labels (+1, −1), through the variational quantum circuit. The classification criterion is based on the parity of the qubits read after the execution of the circuit. If most of the qubits read have an even value, the image is classified as belonging to label 1. If the majority of the qubits have an odd value, the image is classified as belonging to label −1.

The evolution of the cost function during the optimization process with the COBYLA algorithm for 50 iterations in the PD_ARC combination is shown in Figure 9a. Similarly, the evolution of the cost function for the PD_NOPD combination, for 100 iterations, is presented in Figure 9b. It should be noted that, from iteration 50 onwards, no significant improvement in the cost function is observed. In this last combination, an accuracy of 93% can be considered acceptable.

### 3.5. Verification of Results

All experiments performed with quantum computers were performed with the following: IBM Brisbane equipped with the Eagle r3 processor (version 1.1.33), IBM Kyoto, based on the Eagle r3 processor (version 1.2.38), and the IBM Osaka with the Eagle r3 processor (version 1.1.8).

All these systems are based on 127 superconducting qubits [24,28] and use a set of basic logic gates including ECR, ID, RZ, SX, and X (see Section 3.3), with a processing capacity of 5000 CLOPS. The mean errors of the SX and ECR gates, the mean readout error, the average error per logic gate (EPLG), and the system coherence times T1 and T2 for the three computers used are summarized in Table 2.

The first checkpoint is related to the final measurement stage for the selection between one class or another based on the measured parity. Figure 10a presents the measurements obtained for two random images of the PD class, assigned to odd parity. Also shown are the measured results for all combinations, both in the simulation on a classical computer and the results obtained experimentally on the real IBM Osaka quantum computer. Similarly, the measurements obtained for two random images belonging to the NOPD class, with even parity assignment, are presented in Figure 10b. The execution time of the job with 4000 shots on the IBM Osaka computer in both cases is approximately 4 s.

The second verification point consists in obtaining the average accuracy for 136 random images from the test set. The tests are performed for the combinations PD_NOPD, PD_BREAK, PD_ARC, and BREAK_NOPD with the quantum circuits optimized in Section 3.4 by running these circuits on the IBM quantum computers, Kyoto, Brisbane, and Osaka.

The results of all runs are summarized in Table 3. The first four rows represent the average accuracy for the training and test image sets for the simulation and the IBM Kyoto, Brisbane, and Osaka computers. The number of shots is equal to 4096 for each of the quantum circuits. The average execution time for these 136 quantum circuits is 150 s.

In Table 3, it can be observed that the accuracy of the test set of 136 random images for the combination PD_NOPD is 92% on the three quantum computers used, and the error compared to the simulation is 1%. For the rest of the combinations, PD_BREAK, PD_ARC, and BREAK_NOPD, the accuracy for the test set is around 88%.

## 4. Support Vector Machine, SVM

This section describes the basic principles of the second method developed to identify discharges in mineral oils. This method is based on SVM, which has been classically applied to numerous binary classification problems and is the basis of the QKM used in this paper. Section 4.1 describes the SVM primal problem, Section 4.2 explains the advantages of formulating the linear dual problem of SVM, and Section 4.3 generalizes it to a nonlinear SVM problem.

### 4.1. SVM Primal Problem

The primal problem of the SVM is that its dimensionality depends on the number of features *n* in the data. It can be stated as follows [10]: given a training data set {(*x_i_*,*y_i_*)} with features *x_i_* ∈ *ℝn* and labels of a binary classification *y_i_* ∈ {−1,1}, the objective is to find a hyperplane that maximizes the margin between the two classes while allowing certain classification errors.

The primal problem can be formulated as follows:(14)$$\underset{w,b,\xi }{\min}\left(\frac{1}{2}{\left\|w\right\|}^{2}+C_{1}\sum_{i=1}^{m}\xi_{i}^{p}\right)$$
subjected to the following restrictions:(15)$$y_{i}\left(w\cdot x_{i}+b\right)\geq 1-\xi_{i},\;\;\;\;\;\;\xi_{i}\geq 0\;\;\;\;\;para\;i=1,\;2,\cdots ,m$$
where *m* is the number of available samples. For any *p* > 0, it is a convex problem and therefore has a unique solution [8].

In this work, we chose *p* = 2, called the L2 soft margin, which is a common practice, *w* is the weight vector, b is the bias, *ξ_i_* are the slack variables that allow misclassifications, and *C*_1_ is a parameter that controls the trade-off between margin and classification error. A large *C*_1_ results in a smaller margin but fewer misclassifications, while a small *C*_1_ results in a larger margin but more allowed errors.

### 4.2. SVM Linear Dual Problem

The dual formulation is based on the inner products between pairs of data samples, $x_{i}^{T}x_{j}$. These products define the linear kernel *K(x_j_,x_i_)*. The kernel allows these inner products to be computed in the upper feature space efficiently, even when this space is very high-dimensional or infinite.

The dual formulation of the SVM problem [10] with a soft L2 margin is expressed in the following equation:(16)$$\underset{\alpha }{\max}\;\;\;\;\sum_{i=1}^{m}\alpha_{i}-\frac{1}{2}\sum_{i=1}^{m}\sum_{j=1}^{m}\alpha_{i}\alpha_{j}y_{i}y_{j}x_{i}^{T}x_{j}-\frac{1}{4C_{1}}\sum_{i=1}^{m}\alpha_{i}^{2}$$
and is subject to the following restrictions:(17)$$\sum_{i=1}^{m}\alpha_{i}y_{i},\;\;\;\;\;0\leq \alpha_{i}\leq C_{1},\;\;\;\;i=1,\cdots ,m$$

The solution to Equation (16) subjected to the constraints of the dual problem shown in Equation (17) produces the optimal values of *α_i_*. Once we have solved the dual problem and found the *α_i_*, the weight vector *w* can be calculated with the following equation:(18)$$w=\sum_{i=1}^{m}\alpha_{i}y_{i}x_{i}$$

The bias *b* can be calculated using any of the support vectors *x_k_* which are the points for which *α_i_* > 0, as expressed in the following equation:(19)$$b=y_{k}-w^{T}x_{k}$$

Working with the dual problem of an L2 soft margin SVM offers several key advantages.

In the dual problem, the optimization is performed based on the Lagrange multipliers *α_i_*, which are associated with each data sample. This allows the optimization problem to be changed from the feature space, which has a dimension *n*, to the sample space, with a dimension *m*, facilitating data handling.

In the solution of the dual problem, many of the Lagrange multipliers *α_i_* are zero. Only the samples located on the margin, known as support vectors, have *α_i_* > 0. This reduces both the computational cost and the complexity of the model.

Quadratic optimization methods used to solve the dual problem are well developed and especially efficient with high-dimensional problems in the sample space. Specific algorithms, such as the sequential minimal optimization (SMO) method [40], are designed to efficiently solve the SVM dual problem by taking advantage of the sparsity in the Lagrange multipliers.

### 4.3. SVM Dual Nonlinear Problem

If the problem is nonlinear [8], the solution is stated as follows according to Equation (20) subject to the restrictions of Equation (21):(20)$$\underset{\alpha }{\max}\;\;\;\;\sum_{i=1}^{m}\alpha_{i}-\frac{1}{2}\sum_{i=1}^{m}\sum_{j=1}^{m}\alpha_{i}\alpha_{j}y_{i}y_{j}K\left(x_{i},x_{j}\right)-\frac{1}{4C_{1}}\sum_{i=1}^{m}\alpha_{i}^{2}$$(21)$$\;0\leq \alpha_{i}\leq C_{1},\;\;\;\sum_{i=1}^{m}\alpha_{i}y_{i}\;$$

The bias value *b* is calculated using the support vectors *x_i_*, for which the decision function is equal to the label *y_i_* of the following equation:(22)$$y_{i}=\sum_{j=1}^{m}\alpha_{j}y_{j}K\left(x_{j},x_{i}\right)+b$$

To calculate *b*, Equation (18) is rearranged, as indicated below:(23)$$b=y_{i}-\sum_{j=1}^{m}\alpha_{j}y_{j}K\left(x_{j},x_{i}\right)$$
or an average is taken to obtain a more robust estimate of the bias *b*, which is calculated using the following equation:(24)$$b=\frac{1}{\left|S\right|}\sum_{i\epsilon S}\left(y_{i}-\sum_{j=1}^{m}\alpha_{j}y_{j}K\left(x_{j},x_{i}\right)\right)$$
where *S* is the set of support vectors.

Once the value of *b* is determined, the prediction of a new element value *x* is made according to Equation (25):(25)$$y=sign\left(\sum_{j=1}^{m}\alpha_{j}y_{j}K\left(x_{j},x\right)+b\right)$$

That is, we identify the class *y_i_* ∈ {−1,1} to which a new element *x* belongs.

## 5. Quantum Kernel Estimate (Quantum Kernel Model, QKM)

In this section, a quantum kernel estimate is made for two, three, and eight features using a quantum computer with two, three, and eight qubits, respectively. Once the kernel has been estimated, Equations (20)–(25) are used on a classical computer to make the corresponding membership estimate of a new element *x*.

The quantum kernel estimation algorithm consists of mapping classical data vectors into quantum states [11]. This is achieved by a mapping that transforms a classical feature vector x into a quantum state |*ϕ*(*x*)⟩. This process is performed by parameterizing a quantum circuit with the feature x, transforming a unitary matrix over *n* qubits into the ground state |0*^n^*⟩, i.e., *U*(*x*)|0*^n^*⟩.

A quantum kernel is based on using quantum states to represent data and calculate the similarity between them in a quantum feature space. This is done using quantum circuits that encode the data into quantum states. The similarity between two feature vectors *x* and *y* is calculated by the Hilbert–Schmidt inner product between density matrices [8]. The similarity between two feature vectors *x* and *y* is calculated as the value of the quantum kernel *k*(*x*,*y*) given in Equation (26):(26)$$k\left(x,y\right)={\left|\left\langle \phi \left(x\right)|\phi \left(y\right)\right\rangle \right|}^{2}={\left|\left\langle 0^{n}|U^{†}\left(x\right)U\left(y\right)|0^{n}\right\rangle \right|}^{2}$$

The way to evaluate each point in the matrix *k*(*x*,*y*) is to run a quantum circuit *U*^†^(*x*)*U*(*y*) on the input $|\left.0^{n}\right\rangle$ and find the probability of obtaining the state $|\left.0^{n}\right\rangle$. Figure 11 shows the generic structure of the quantum circuit used to estimate the particularized kernel for three features. For this purpose, the ZZFeatureMap function obtained from the Qiskit library [28] was used. This is a parameterized quantum circuit used to map classical data to a quantum feature space. This mapping is performed by applying quantum rotation gates on the qubits, followed by CX-type interactions between pairs of qubits, which allows the capture of nonlinear relationships in the data.

Next, in Section 5.1, the results obtained with the QKM method for the binary combinations PD_NODP, PD_BREAK, PD_ARC, and BREAK_NOPD, with two features, are explained in order to make a comparison with the results obtained with the QVM method. Then, in Section 5.2, the results with the QKM method for three and eight features are analyzed, and the improvement in accuracy is analyzed.

### 5.1. Two-Feature Kernel Estimation

All simulations performed on classical computers and experiments performed on quantum computers were run following the basic ZZFeatureMap circuit structure, shown in Figure 11 and defined in [35]. In this section, the kernel is estimated for two features, [‘area-pixels’, ‘mean-coords-x’], described in Section 2 and for the four class combinations PD_NODP, PD_BREAK, PD_ARC, and BREAK_NOPD.

The exact kernel results for the PD_BREAK class combination are depicted in Figure 12a. Only the upper diagonal needs to be computed, since the matrix is symmetric. The element kernel matrix obtained is 580 × 580, as can be seen in the Data Index (X1) and (X2). The number of computations, which corresponds to the quantum circuit in Figure 11, is determined by Equation (26). This circuit was used to estimate the kernel on a quantum computer, performing 168,200 evaluations. For each evaluation, the kernel matrix value is normalized between 0 and 1.

Figure 12b shows a comparative study between the exact kernel value and the value obtained with the IBM Kyoto computer for row 40 and columns 0 to 24 (Pub 0 to Pub 24) of the matrix shown in Figure 12a. The matrix was generated using the library Qiskit [28], with the Jupyter Notebook QKM_verification_two_qubits.ipynb allocated in the repository [29].

Due to time constraints on available quantum computers, 140 values were randomly selected from the top of the matrix to estimate the kernel and compared with those obtained in simulations on a classical computer. These values are shown in Figure 13a, with the execution time on the quantum computer being 2 m 34 s. The results of the simulation and the execution on the IBM Kyoto computer are presented in Figure 13b, with a mean absolute percentage error (MAPE) of 7.6% according to Equation (27). Figure 12b details the comparison between the exact value and the results of 25 consecutive values from row 40 of the kernel matrix, obtained on the same quantum computer.(27)$$MAPE=\frac{100}{k}\sum_{i=1}^{k}\left|\frac{A_{i}-F_{i}}{A_{i}}\right|$$
where *k* = 140 is the total number of elements, *A_i_* is the actual observed value, and *F_i_* is the value predicted by the simulation.

Figure 14a shows another 140 random evaluations, different from the previous ones, see, performed on another quantum computer, IBM Osaka, for PD_BREAK class combination, in order to verify the results in another physical environment. Figure 14b shows the comparison between the results obtained on the real IBM Osaka quantum computer and the simulation, with a runtime of 2 m 34 s on the IBM computer.

The results obtained in the three quantum computers were verified in the other three class combinations PD_NOPD, BREAK_NOPD, and PD_ARC, and are represented in Figure 15, Figure 16, and Figure 17, respectively, along with the simulations of the kernel obtained with a classical computer.

The exact kernel results for the PD_NOPD, BREAK_NOPD, and PD_ARC class combinations are depicted in Figure 15a, Figure 16a, and Figure 17a, respectively. Only the upper diagonal needs to be computed, since the matrix is symmetric. The element kernel matrix obtained is 580 × 580, as can be seen in the Data Index (X1) and (X2). The number of computations, which corresponds to the quantum circuit in Figure 11, is determined by Equation (26). This circuit was used to estimate the kernel on a quantum computer, performing 168,200 evaluations. For each evaluation, the kernel matrix value is normalized between 0 and 1.

Figure 15b, Figure 16b, and Figure 17b show a comparative study between the exact kernel value and the value obtained with IBM Osaka, Brisbane, and IBM Kyoto computers, respectively, for row 40 and columns 0 to 24 (Pub 0 to Pub 24) of the matrix shown in Figure 15a, Figure 16a, and Figure 17a, respectively. These matrices were generated using the library Qiskit [28], with the Jupyter Notebook QKM_verification_two_qubits.ipynb allocated in the repository [29].

As a final summary of the results obtained for two features, Table 4 presents the accuracy of all the binary combinations using quantum kernel estimation on the different two-qubit features x = [‘area-pixels’, ‘mean-coords-x’], using a test image ratio of 20%. The accuracy values are 83% and 97% for the PD_ARC and PD_NOPD combinations, respectively. The execution times for the kernel calculation on the test items vary between 2587 s and 2639 s for these models.

Results for the QVM are presented in Table 3 (see Section 3). In the first QVM, an accuracy of 92% was observed on the test set for 136 random images under the first PD_NOPD binary combination, obtained consistently on the three types of quantum computers used, with a 1% error margin compared to the simulation. For the other three binary combinations, the test set accuracy was around 88%.

However, for QKM, using SVM, the test set accuracy for the PD_NOPD combination reached 97%, while for the other combinations, the average accuracy was 89% (Table 4).

### 5.2. Kernel Estimation with Three and Eight Features

In this section, the improvement in accuracy when increasing the number of features is analyzed.

The features used for three qubits are represented in Equation (28). The test or validation set represents 80% of the data.(28)$$X=\left[\begin{array}{c} area\_pixels\\ mean\_coords\_x\\ mean\_coords\_y \end{array}\right]$$

Table 5 shows the results of four binary combinations, using an SVM with a kernel to train and evaluate models with these three features, parameter *C*_1_ = 1 in Equation (20). The results indicate that the accuracies on the test set for the binary combinations PD_NODP, PD_BREAK, PD_ARC, and BREAK_NOPD are 99%, 85%, 84%, and 92%, respectively. The kernel matrix computation times for the training data vary between 157 s and 307 s, corresponding to a range of 129 to 173 evaluations. For the validation set, the times range between 1290 and 3509 s, due to a higher number of evaluations.

The features used for eight qubits are expressed in Equation (29). The test or validation set represents 80% of the data.(29)$$X=\left[\begin{array}{c} area\_pixels\\ mean\_coords\_x\\ mean\_coords\_y\\ centroid\_x\\ centroid\_y\\ meanintensity\\ stdintensity\\ threshold \end{array}\right]$$

Table 6 shows the results of four binary combinations, using an SVM with a kernel to train and evaluate models with these eight features. The results indicate that the accuracy on the test set for all the binary classes PD_NODP, PD_BREAK, PD_ARC, and BREAK_NOPD is 99%, 94%, 99%, and 98%, respectively. The kernel matrix computation times for the training data vary between 294 and 531 s, corresponding to a range of 129 to 173 evaluations. For the validation set, the times range between 2352 s and 4229 s, due to a higher number of evaluations.

A comparison of Table 5 and Table 6 shows how increasing the number of features affects both the accuracy and the execution time of the SVM models with the kernel. When using more features (eight instead of three), an improvement in accuracy is observed, especially on the test set, suggesting that the model generalizes better with new data. However, this improvement leads to an increase in execution times as more features are added.

## 6. Overall Flowchart

In this section, an overall flowchart, see Figure 18, that explains the main steps followed in the article to obtain the different results shown in Section 2, Section 3, Section 4 and Section 5 has been added. We believe this will make the procedure easier to understand.

A detailed description of each step of the process is included in the README of the Zenodo repository [29], along with links to the corresponding Jupyter Notebooks. To make it easier to understand the overall workflow, we have included a summary description of the flowchart in four steps. This flowchart is described below.

Flowchart Description:

The flowchart is generated running the Binary_features_generation.ipynb file and can be viewed directly in the repository. It graphically represents the four main steps of this experimental process.

Step A: Feature Generation

The name of the Jupyter Notebook is Binary_features_generation.ipynb.

It presents the following subsections: image visualization, feature extraction, and binary concatenation of files in csv format. Folders /IMAGES/ and /FEATURE_RESULTS/ are referenced, and the feature_*.csv files are introduced.

Step B: Optimization of QVM Parameters

The name of the Jupyter Notebook is FIT_DP_NODP_CIRCUIT.ipynb.

It presents the following subsections: library import, function definition, data loading, normalization, quantum circuit definition, cost function, COBYLA optimization, and model evaluation. Reference is made to Table 1 and to the optimal parameters stored in the variable named opt_var.

Step C: Verification on Quantum Hardware/Simulation

The name of the Jupyter Notebook is QVM_verification_two_qubits.ipynb.

It presents the following subsections: environment setup, backend selection, data and parameter loading, circuit definition, transpilation, circuit execution, and results analysis. Reference is made to Table 3 and the results comparison graphs.

Step D: Execution of QKM Model

The name of the Jupyter Notebook is QKM_verification_two_qubits.ipynb.

It presents the following subsections: execution of the QKM model on real quantum computers, real quantum computer execution, quantum kernel estimation algorithm, and SVM with QKM for two, three, and eight qubits.

## 7. Conclusions

In this paper, electrical discharge images are classified using AI with quantum machine learning techniques. The results show that quantum machine learning is effective in classifying electrical discharge images in dielectric mineral oils that were detected by a high-resolution optical sensor. Both the variational quantum model QVM and the support vector machine SVM with quantum kernel model estimation QKM achieved significant accuracies of 92% and 97%, respectively, in the first discharge combination, PD_NOPD, realized with two qubits. This demonstrates the potential of quantum algorithms in classification applications in highly complex scenarios.

The two developed quantum models showed remarkable consistency when running on three different physical quantum computers, IBM Osaka, IBM Brisbane, and IBM Kyoto. The results obtained a 1% error margin compared to simulations performed on classical computers, indicating the robustness of the models against variability in quantum hardware. Increasing the number of qubits from two to eight in the QKM resulted in a significant improvement in model accuracy, reaching an average of 97% in test set accuracy for the four binary combinations PD_NOPD, PD_BREAK, PD_ARC, and BREAK_NOPD. This increase shows that models with more qubits have a higher generalization capacity, improving the classification of previously unseen data.

The comparison between the two models indicates that although both quantum approaches proved to be effective, the SVM-QKM slightly outperformed the QVM in terms of overall accuracy. This result suggests that, in this specific context, combining quantum techniques with classical methods such as SVM can offer an advantage in classifying complex patterns.

The findings show that implementing quantum machine learning techniques in detection and diagnostic systems offers significant advantages in terms of accuracy and generalization capacity, especially when more qubits are employed in the models. This opens the door to future research and practical applications in the field of detection and analysis of electrical discharges in dielectric systems.

## Figures and Tables

**Figure 1 sensors-25-01277-f001:**
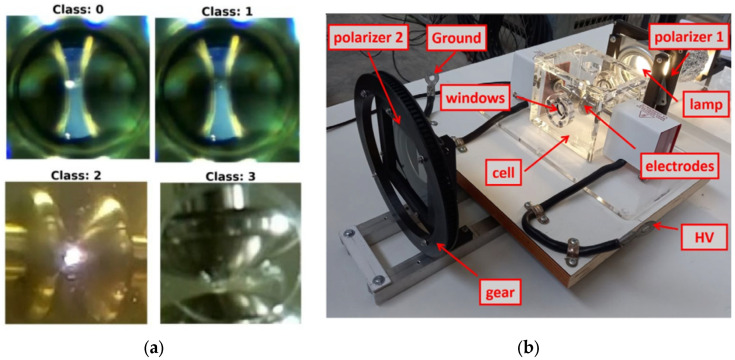
(**a**) Reference images at the electrodes with their four possible classes associated with the discharges in mineral oil, class 0 (PD), class 1 (NOPD), class 2 (ARC) and class 3 (BREAK). (**b**) Experimental device made for the collection of images [31].

**Figure 2 sensors-25-01277-f002:**
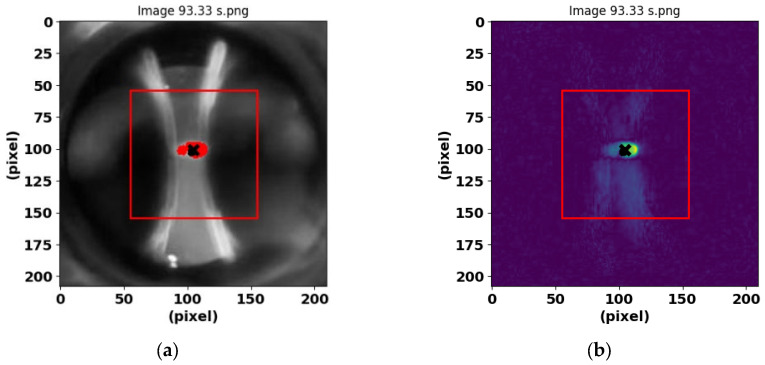
PD and its processing at 93.33 s. PD are shown in red and green color. (**a**) Selection and analysis of ROI, electrodes, and class 0 discharge. (**b**) Image obtained with the Scikit-image environment [27].

**Figure 3 sensors-25-01277-f003:**
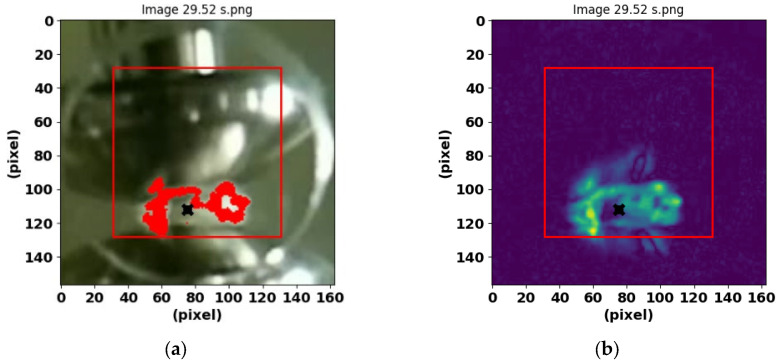
Features obtained from the BREAK class are shown in red and green. (**a**) Image of the electrodes and the red-colored area of the post-arc bubbles, obtained at time 29.52 s. (**b**) Image resulting after the Scikit-image program was applied.

**Figure 4 sensors-25-01277-f004:**
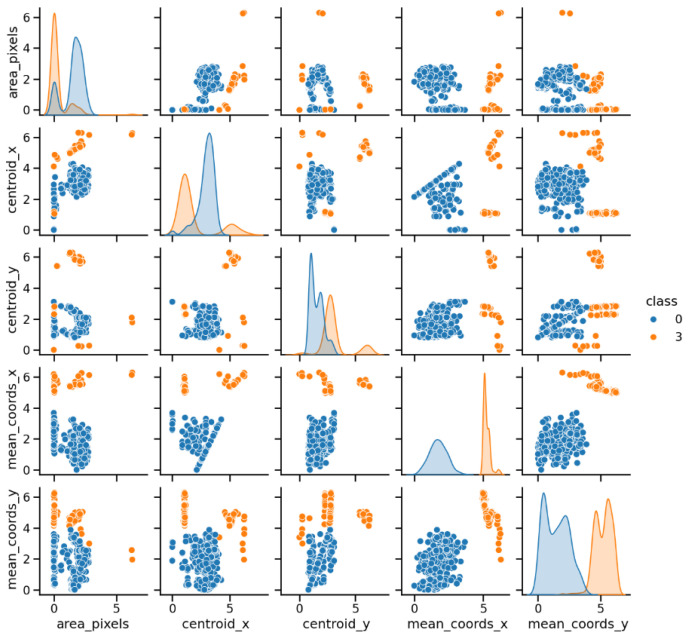
Relationships between five of the thirteen normalized features in the interval [0, 2*π*] corresponding to the PD-BREAK class combination.

**Figure 5 sensors-25-01277-f005:**
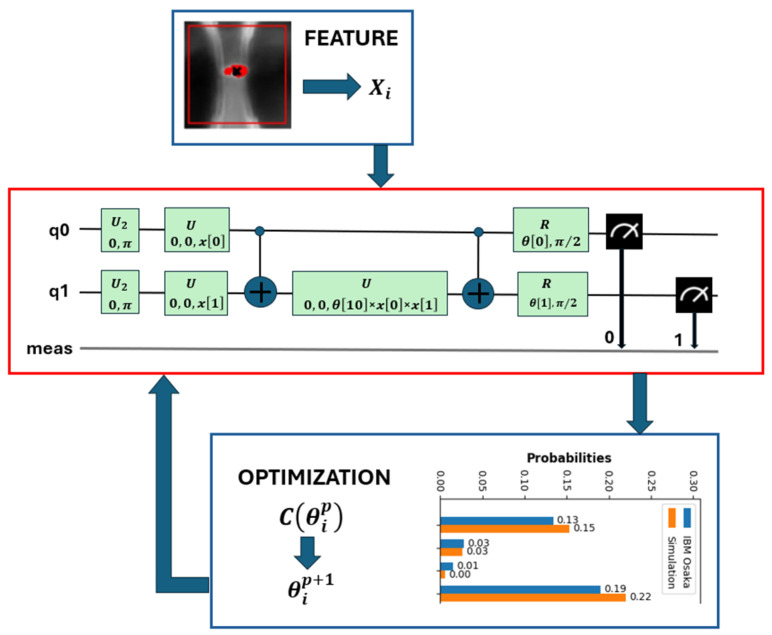
Main blocks of a QVM. The red block runs on the quantum computer, the blue one on the classical.

**Figure 6 sensors-25-01277-f006:**
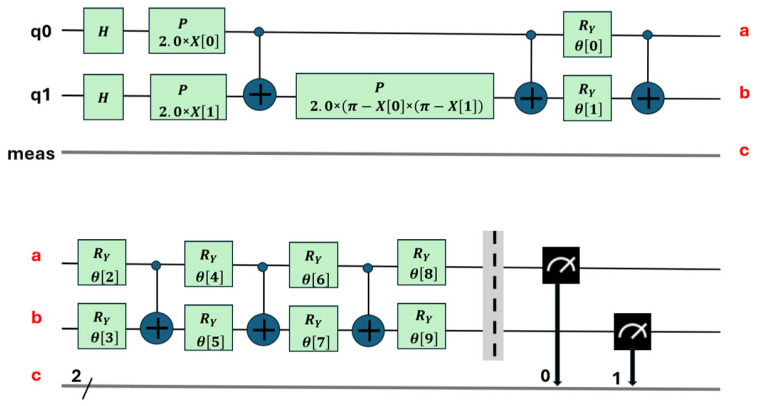
*H* is the Hadamard gate presented in Equation (7); *P* is the phase gate that performs a rotation around the z axis of the complex plane; *Ry* performs a rotation around the *y* axis. This circuit contains 10 parameters. a, b and c show the logical connection of the two levels in the diagram.

**Figure 7 sensors-25-01277-f007:**
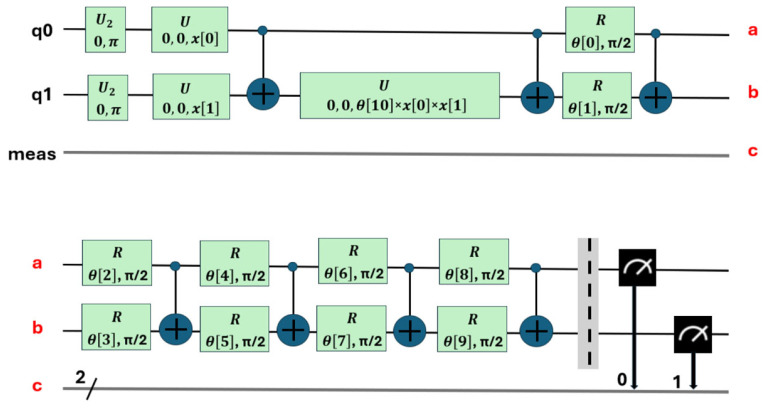
*U*_2_ = *U*_2_(*ϕ*,*λ*) = *U*_2_(0,*π*) = *H*, *U*(*θ*,*ϕ*,*λ*) = *U*(0,0,*λ*) = *P*(*λ*); *R* = *Ry*. This circuit contains 11 parameters. a, b and c show the logical connection of the two levels in the diagram.

**Figure 8 sensors-25-01277-f008:**
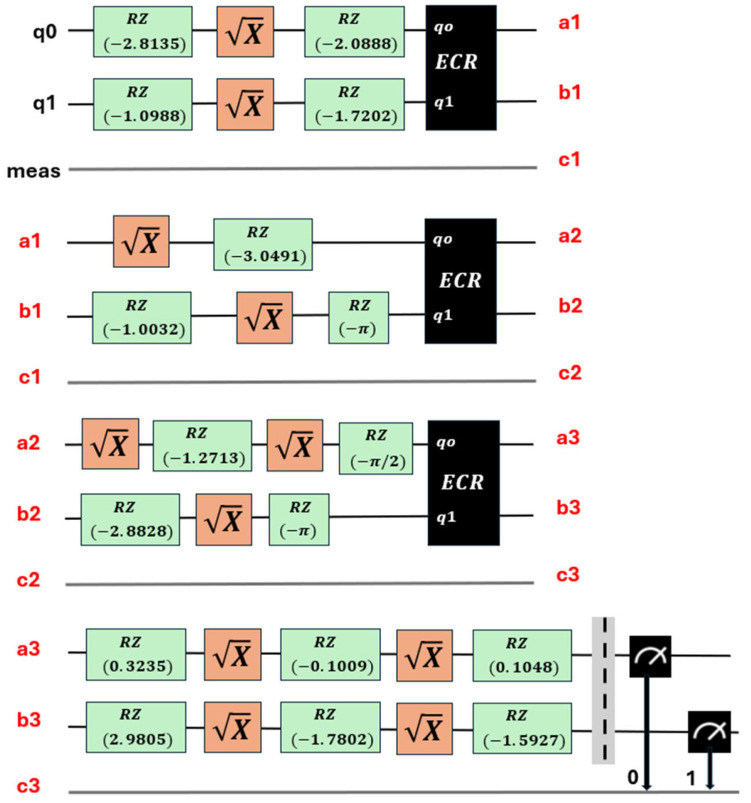
Transpilation of the 11 parameters of the circuit shown in Figure 7, where a1–a3, b1–b3, c1–c3 show the logical connection of the four levels in the diagram.

**Figure 9 sensors-25-01277-f009:**
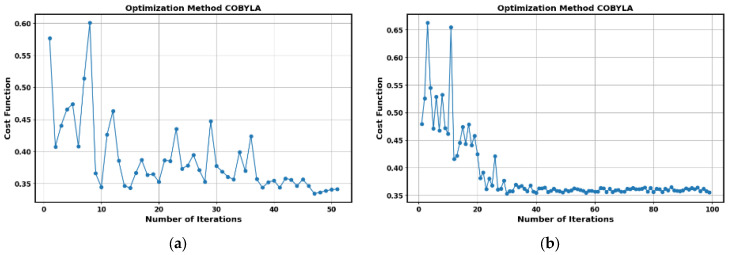
Evolution of the cost function in the optimization process for the COBYLA algorithm corresponding to Table 1. (**a**) Evolution of the cost function for the PD_ARC combination. (**b**) Evolution of the cost function for the PD_NOPD combination.

**Figure 10 sensors-25-01277-f010:**
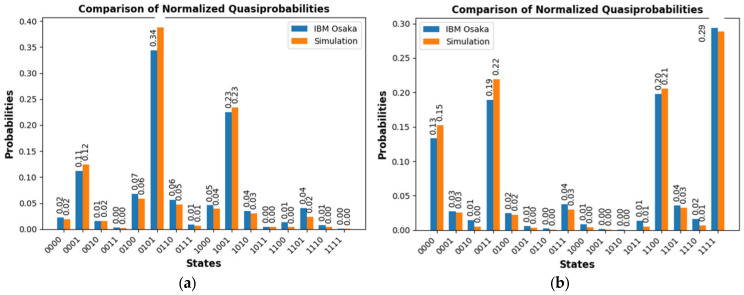
Measurement results for the PD_NOPD model for two images with the IBM Osaka quantum computer vs. simulation. (**a**) Measurements obtained for two values belonging to the PD class, odd parity. (**b**) Measurements obtained for two values belonging to the NOPD class, even parity.

**Figure 11 sensors-25-01277-f011:**
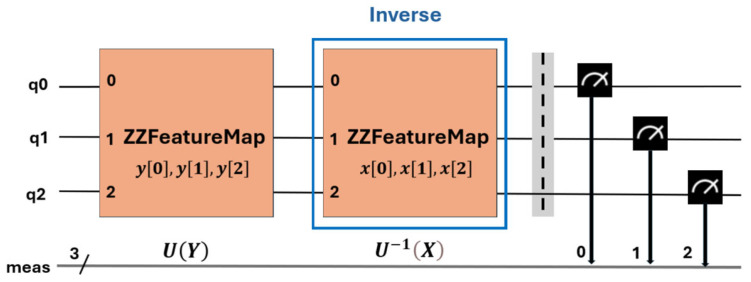
Generic structure of the quantum circuit and measurement used to estimate the kernel of Equation (26). It is particularized for three features. The number 3 represents the set of the three measured features 0, 1 and 2. q0, q1 and q2 are the input qubits to the quantum circuit.

**Figure 12 sensors-25-01277-f012:**
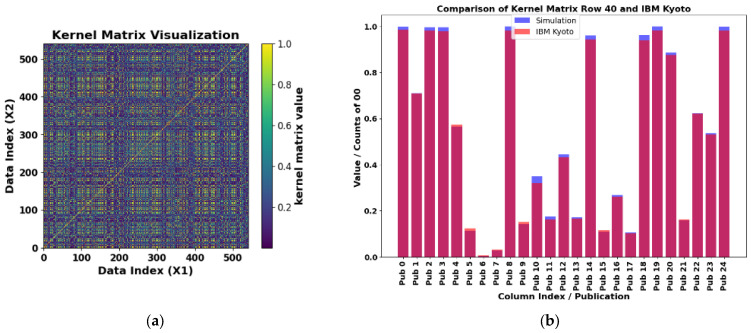
(**a**) Exactly computed two-feature kernel matrix for the PD_BREAK combination. Simulation result of the symmetric 580 × 580 element kernel matrix obtained with Equation (26), for PD_BREAK. (**b**) Comparison of results for the IBM Kyoto computer. Verification of results for row 40 and columns 0 to 24 of the matrix shown in (**a**), simulation in blue. Where the orange and blue colors overlap, a magenta color is displayed.

**Figure 13 sensors-25-01277-f013:**
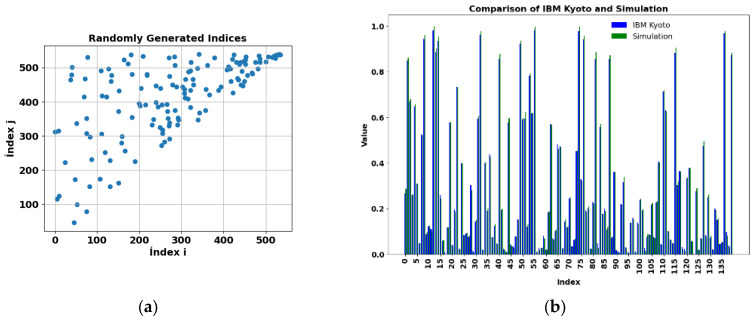
(**a**) One hundred forty values randomly chosen from the kernel matrix for PD_BREAK. (**b**) Comparison of results for the real IBM Kyoto computer with the simulation and mean absolute percentage error (MAPE) = 7.6%, on the real IBM Kyoto computer.

**Figure 14 sensors-25-01277-f014:**
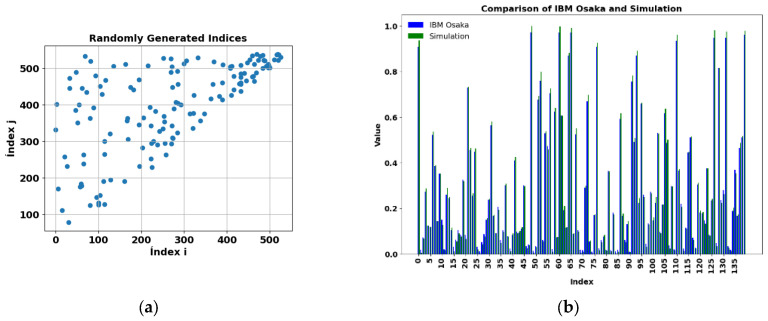
(**a**) One hundred forty randomly selected values from the kernel matrix for PD_BREAK, obtained using the IBM Osaka computer. (**b**) Comparison of results for the real IBM Osaka computer with the simulation, for the PD_BREAK combination. Execution time 2 m 34 s.

**Figure 15 sensors-25-01277-f015:**
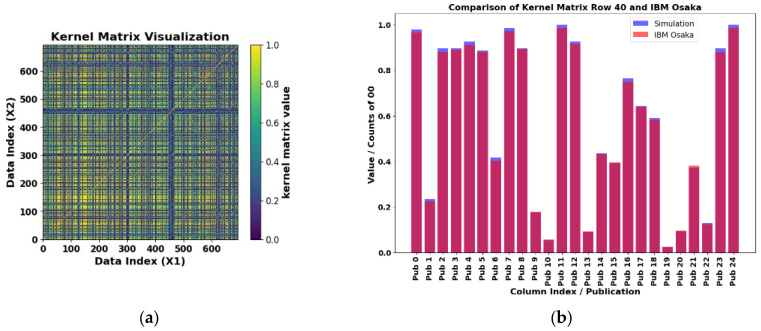
(**a**) Exactly computed two-feature kernel matrix for the PD_NOPD combination. The kernel matrix is symmetrical with 693 × 693 elements. (**b**) Comparison of results for the real IBM Osaka computer and verification of results for row 40 and columns 0 to 24 of the matrix shown in (**a**), simulation in blue. Where the orange and blue colors overlap, a magenta color is displayed.

**Figure 16 sensors-25-01277-f016:**
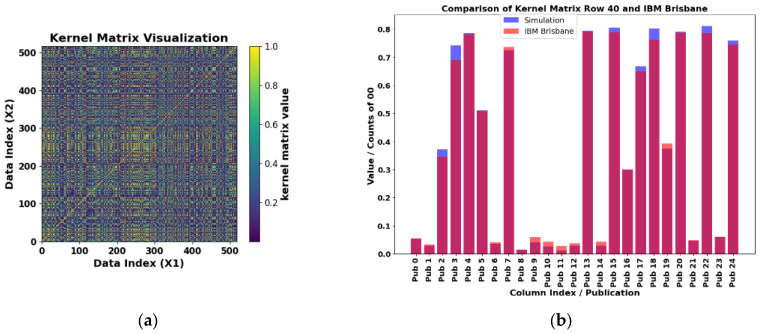
(**a**) Exactly computed two-feature kernel matrix for the BREAK_NOPD combination. The kernel matrix is symmetrical with 517 × 517 elements. (**b**) Comparison of results for the real IBM Brisbane computer and verification of results for row 40 and columns 0 to 24 of the matrix shown in (**a**), simulation in blue. Where the orange and blue colors overlap, a magenta color is displayed.

**Figure 17 sensors-25-01277-f017:**
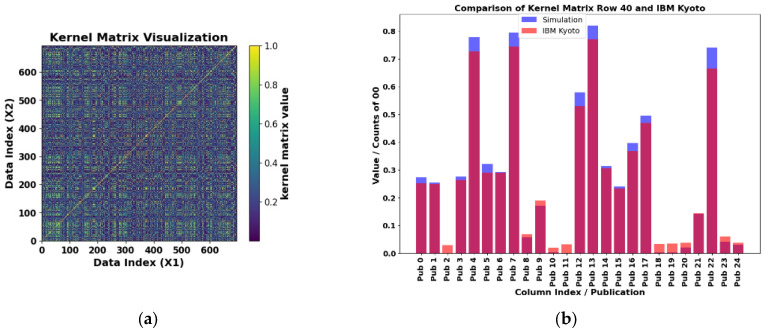
(**a**) Exactly computed two-feature kernel matrix for the PD_ARC combination. The kernel matrix is symmetrical with 694 × 694 elements. (**b**) Comparison of results for the real IBM Kyoto computer and verification of results for row 40 and columns 0 to 24 of the matrix shown in (**a**), simulation in blue. Where the orange and blue colors overlap, a magenta color is displayed.

**Figure 18 sensors-25-01277-f018:**
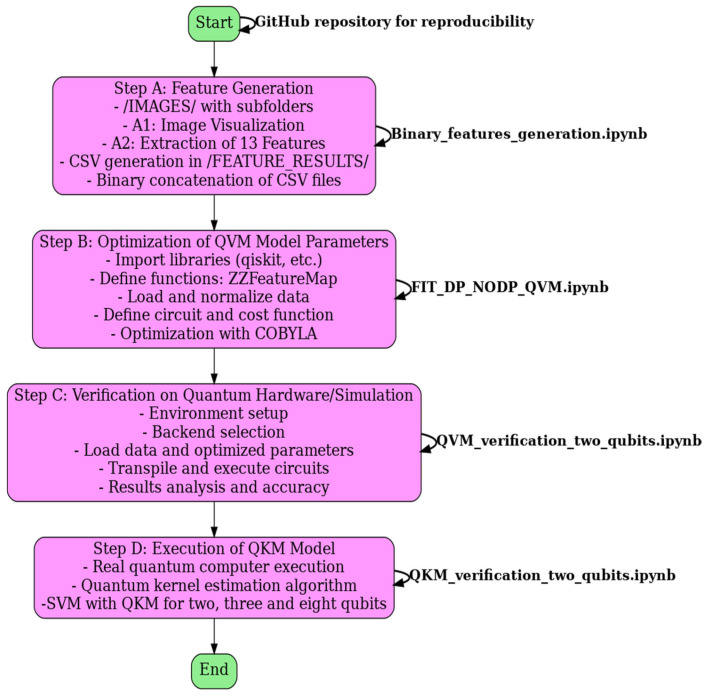
Overall flowchart and Jupyter Notebooks.

**Table 1 sensors-25-01277-t001:** Optimal parameters for the QVM circuits of Figure 6 and Figure 7, with 10 and 11 parameters, respectively.

Optimal Parameters	PD_NOPD	PD_BREAK	PD_ARC	BREAK_NOPD
**θ(0)**	−1.36	−0.46	−2.03	−0.11
**θ(1)**	0.73	3.01	2.49	5.64
**θ(2)**	0.46	0.35	−0.12	−2.54
**θ(3)**	−0.23	−0.43	−0.75	−2.71
**θ(4)**	0.72	−3.04	−1.36	−6.40
**θ(5)**	−5.04	−5.67	−5.43	−1.20
**θ(6)**	0.25	1.70	1.55	−0.99
**θ(7)**	3.20	6.01	5.31	8.97
**θ(8)**	2.22	5.84	5.63	4.15
**θ(9)**	2.8	2.26	3.50	8.22
**θ(10)**	—	1.56	1.18	1.53
**Accuracy** ^1^	[0.90:0.93]	[0.93:0.95]	[0.82:0.83]	[0.85:0.82]

^1^ [train:test].

**Table 2 sensors-25-01277-t002:** Calibration data for computers used in the experiments from IBM.

Mean Property	IBM Osaka	IBM Kyoto	IBM Brisbane
**T1 (µs)**	287.09	215.43	228.55
**T2 (µs)**	144.57	109.44	151.41
**SX error %**	3.053 × 10^−2^	3.073 × 10^−2^	2.409 × 10^−2^
**ECR error %**	8.032 × 10^−1^	9.345 × 10^−1^	7.820 × 10^−1^
**EPLG error %**	3.3	3.6	2.0
**Readout error %**	2.210	1.540	1.350

**Table 3 sensors-25-01277-t003:** Accuracy, time (s), and numbers of variational quantum circuits used in the IBM computers in the experiments corresponding to Figure 6 and Figure 7, with 10 and 11 parameters, respectively.

	PD_NOPD	PD_BREAK	PD_ARC	BREAK_NOPD
Accuracy ^1^ simulation	[0.90:0.93]	[0.93:0.95]	[0.82:0.83]	[0.85:0.82]
Accuracy ^1^ Kyoto	[0.90:0.92]	[0.95:0.91]	[0.80:0.88]	[0.85:0.87]
Accuracy ^1^ Brisbane	[0.90:0.92]	[0.93:0.91]	[0.80:0.88]	[0.84:0.85]
Accuracy ^1^ Osaka	[0.92:0.92]	[0.95:0.91]	[0.80:0.80]	[0.85:0.88]
Time (s) ^1^ Kyoto	[150:150]	[150:150]	[150:150]	[149:110]
Time (s) ^1^ Brisbane	[151:150]	[110:149]	[149:150]	[149:110]
Time (s) ^1^ Osaka	[150:150]	[149:149]	[149:149]	[149:110]
Circuits ^1^ Kyoto	[136:136]	[136:136]	[136:136]	[136:100]
Circuits ^1^ Brisbane	[136:136]	[100:136]	[136:136]	[136:100]
Circuits ^1^ Osaka	[136:136]	[136:136]	[136:136]	[136:100]

^1^ [train:test].

**Table 4 sensors-25-01277-t004:** Accuracy and execution times for two qubits x = [‘area-pixels’, ‘mean-coords-x’] with test_size = 20%.

	PD_NOPD	PD_BREAK	PD_ARC	BREAK_NOPD
**Accuracy [train]**	0.95	0.90	0.84	0.93
**Accuracy [test]**	0.97	0.94	0.83	0.92
**Train execution time (s)**	5040	3573	5357	2911
**Test execution time (s)**	2587	1768	2639	1476
**SVM fit training time (s)**	0.019	0.016	0.018	0.012
**Matrix dimension**	693	580	694	517
** *C* _1_ **	1	1	1	1

**Table 5 sensors-25-01277-t005:** Accuracy and execution times for three qubits Equation (28), with symmetric matrix, test_size = 80%.

	PD_NOPD	PD_BREAK	PD_ARC	BREAK_NOPD
**Accuracy [train]**	0.99	0.95	0.92	0.95
**Accuracy [test]**	0.99	0.85	0.84	0.92
**Train execution time (s)**	307	177	307	157
**Test execution time (s)**	3509	1300	3509	1290
**SVM fit training time (s)**	0.004	0.001	0.004	0.001
**Matrix dimension**	173	135	173	129
** *C* _1_ **	1	1	1	1

**Table 6 sensors-25-01277-t006:** Accuracy and execution times for eight qubits Equation (29) with symmetric matrix, test_size = 80%.

	PD_NOPD	PD_BREAK	PD_ARC	BREAK_NOPD
**Accuracy [train]**	1	1	1	1
**Accuracy [test]**	0.99	0.94	0.99	0.98
**Train execution time (s)**	531	323	529	294
**Test execution time (s)**	4229	2567	4214	2352
**SVM fit training time (s)**	0.001	0.001	0.001	0.001
**Matrix dimension**	173	135	173	129
** *C* _1_ **	1	1	1	1

## Data Availability

Data are contained within the article.

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
