# Peer review of "Quantum Variational vs. Quantum Kernel Machine Learning Models for Partial Discharge Classification in Dielectric Oils"

_sensors, 2025, doi:10.3390/s25041277_

Round 1
Reviewer 1 Report
Comments and Suggestions for Authors
In this work, a quantum kernel machine learning model is proposed for the partial discharge classification of dielectric oils. Many results have been presented, but the contribution of this work is not clear. Following are the main comments:
1. As for the abstract and introduction, the authors should pay attention to highlighting their contribution and novelty compared with the existing models. Especially the introduction, this part of the content should be re-organized to highlight the confronting challenge and corresponding measures to solve these questions. Regarding this, the novelty of this work is not clear.
2. Contained figures are too obscured since all of the used data are not collected by the authors. A big enough picture database is so fundamental for training the classifier for PD diagnosis, how do authors complete this task without the available experimental platform?
3. As for section 2, a flow diagram is good enough to describe this process procedure rather than using such a complex description.
4. The author implemented fault diagnosis based on optical partial discharge spectra, but in practice, how can the discharge spectra of transformer oil gaps be obtained? Therefore, this method is difficult to apply to on-site transformer fault diagnosis.
Author Response
REVIEWER 1
Thank you very much for your time and valuable feedback on our work "Quantum Variational vs Quantum Kernel Machine Learning Models for Partial Discharge Classification in Dielectric Oils". We have carefully reviewed your suggestions and responded to each of the proposed points. Furthermore, for clarity and reproducibility, we have created a public repository on Zenodo containing all the data we have generated for this paper, the source code developed in Python, and a detailed README explaining step by step the experimentation process.
Currently private link to the Zenodo repository:
[29]
https://zenodo.org/records/14589724?preview=1&token=eyJhbGciOiJIUzUxMiJ9.eyJpZCI6IjcxYjY0NmJkLTVhN2MtNGExZC1iMTM1LTRkYzJkZDBhODlkNCIsImRhdGEiOnt9LCJyYW5kb20iOiIwNTg5ZmIzODg4ODMzODY3N2IxYmZiNGRmZGJkODJlNCJ9.Cuy1bJrWskY0Z0ZeQkJcVyx05hipFSLE797lYY-y7hI2GTBir1dIXhlDX3zIeEYYaVO2pFaE8nVYT2poUZnF8w
Reference to the final public repository in Zenodo:
[29] Monzón-Verona, J. M., García-Alonso, S., & Santana-Martín, F. J. 2024. Software and dataset of Quantum Variational vs Quantum Kernel Machine Learning Models for Partial Discharge Classification in Dielectric Oils. Zenodo. https://doi.org/10.5281/zenodo.14589724
1. Contribution and novelty of the work
REVIEWER: As for the abstract and introduction, the authors should pay attention to highlighting their contribution and novelty compared with the existing models. Especially the introduction, this part of the content should be re-organized to highlight the confronting challenge and corresponding measures to solve these questions. Regarding this, the novelty of this work is not clear.
ANSWER:
We appreciate the reviewer's comment.
We have revised and modified both the abstract and the introduction to more clearly emphasize the contribution and novelty of our work.
- Main contribution: Our work demonstrates, for the first time, the application and comparison of two quantum machine learning models (QVM and QKM) for the classification of partial discharge (PD) images in dielectric oils, using real data obtained with a high-resolution optical sensor. Furthermore, the impact of increasing the number of qubits on the accuracy of the QKM model is explored.
• Novelty:
- Application to a real problem: unlike previous studies that focus on simulated or theoretical data sets, our work is applied to a real and practical problem in the field of dielectric oil diagnostics of transformers, using real PD images.
- Comparison of QVM and QKM: a comprehensive comparison is made between the QVM and QKM models in the specific context of PD classification, showing the advantages and limitations of each.
- Study of the impact of the number of qubits in QKM: It is shown that increasing the number of qubits in the QKM model significantly improves the classification accuracy of the four binary combinations of PD classes.
- Using real quantum computers: the models were implemented and run on three IBM fault-tolerant quantum computers, demonstrating their operation on real quantum hardware and providing results comparable to classical simulations.
- Transparency and reproducibility: a repository has been created on Zenodo, with a detailed README, where the images of the electric discharges used, the Jupiter Notebooks for the extraction of the features, and the Jupyter Notebooks of the QVM and QKM have been published, all in order to facilitate the transparency and reproducibility of our results, so that the scientific community can access and use them.
In-situ experimental images of transformer oil spaces are extremely complex. In this work, the tests have been performed in the laboratory with oil samples extracted from the transformer.
Consider that the main objective of this work is to explore the feasibility and potential of quantum machine learning models for the classification of electrical discharges in dielectric oils, using a controlled laboratory environment.
Although our current work is not focused on in-situ studies, it lays the groundwork for future research in that direction. This laboratory set-up offers the advantage of immunity to electromagnetic interference.
Therefore, on line 43 we have introduced the following text:
“In-situ experimental images of transformer oil spaces are extremely complex. In this work, the tests have been performed in the laboratory with oil samples extracted from the transformer. Consider that the main objective of this work is to explore the feasibility and potential of quantum machine learning models for the classification of electrical discharges in dielectric oils, using a controlled laboratory environment.
Although our current work is not focused on in-situ studies, it could lay the groundwork for future research in that direction. This laboratory set-up offers the advantage of immunity to electromagnetic interference.”
In addition, on line 183 we have introduced the following text:
“The main contribution of this paper is that for the first time two quantum machine learning models, QVM and QKM, are applied and compared for the classification of electrical discharge images in dielectric oils, using real data obtained with a high-resolution optical sensor.
The novelty of this work can be summarized in the following points. The work is applied to a real problem, unlike other previous studies that focus on simulated or theoretical data sets. In addition, a study is carried out on the impact of the number of qubits in QKM and it is shown that increasing the number of qubits in this model significantly improves the accuracy in the classification of the four binary combinations of the classes. On the other hand, real quantum computers are used and the models are implemented and executed on three fault-tolerant IBM quantum computers, demonstrating their operation on real quantum hardware and providing results comparable to classical simulations. This work also provides transparency and reproducibility by creating a repository on Zenodo [29], with a detailed README, where the images of the electrical discharges used, the Jupyter Notebooks for the extraction of the features, and the Jupiter Notebooks with the Python programming of QVM and QKM with the respective figures have been published, so that the scientific community can access and use them.”
Taking into account the clarifications introduced, the abstract has been modified as follows:
“Abstract: In this paper, electrical discharge images are classified using AI with quantum machine learning techniques. These discharges are originated in dielectric mineral oils and were detected by a high-resolution optical sensor. Two quantum binary classification models were developed and compared for four discharge binary combinations. The first was a quantum variational model (QVM), and the second a conventional support vector machine (SVM) with a quantum kernel model (QKM). The execution of these two models was realized on three fault-tolerant physical quantum IBM computers. The novelty of this article lies in its application to a real problem, unlike other studies that focus on simulated or theoretical data sets. In addition, a study is carried out on the impact of the number of qubits in QKM and it is shown that increasing the number of qubits in this model significantly improves the accuracy in the classification of the four binary combinations studied. In the QVM, with two qubits, an accuracy of 92% was observed in the first discharge combination in the three quantum computers used, with a margin of error of 1% compared to the simulation obtained on classical computers.”
- Figures and image database
REVIEWER: Contained figures are too obscured since all of the used data are not collected by the authors. A big enough picture database is so fundamental for training the classifier for PD diagnosis, how do authors complete this task without the available experimental platform?
ANSWER:
- Data source: all images of the electrical discharges were collected directly by the authors and come from a recently published previous work [31] performed by our research team, where a high-resolution optical sensor was used to capture PD images in a controlled laboratory environment validated with a standard electrical detection system (according to IEC60270). We have clarified this point in the "Materials and Methods" section. From these images, new images have been generated from which the features have been extracted to train the quantum circuits. Within the Zenodo repository in the IMAGES folder there are four subfolders containing all the images of the electrical discharges used.
- Clarity of figures: we recognize that some figures may seem obscure. To facilitate their understanding, access to a public Zenodo database with a more detailed explanation has been added [29], where links to the Jupiter Notebooks used to generate those figures are indicated.
- Size of the original image database: as can be seen in [29], a large database was available, and a sufficient dataset was used to demonstrate the feasibility of the proposed quantum models. The repository we have created allows other researchers to supply more images in the future, thus expanding the database. In addition, all the Jupyter Notebooks used have been provided. They are inside the repository in the root folder in the files Binary_features_generation.ipynb, FIT_DP_NODP_CIRCUIT.ipynb, QKM_verification_two_qubits.ipynb, and QVM_verification_two_qubits.ipynb.
- Overall Flowchart
REVIEWER: As for section 2, a flow diagram is good enough to describe this process procedure rather than using such a complex description.
ANSWER:
We appreciate your suggestion.
A new section 6 has been added to explain the overall flowchart that explains the main steps followed in the article to obtain the different results shown in sections 2 to 5.
The following explanation has been added to the article on line 723:
“6. Overall flowchart
In this section an overall flowchart, see Figure 18, that explains the main steps followed in the article to obtain the different results shown in sections 2 to 5 has been added. We believe this will make the procedure easier to understand.
A detailed description of each step of the process is included in the README of the Zenodo repository [29], along with links to the corresponding Jupyter Notebooks. To make it easier to understand the overall workflow, we have included a summary description of the flowchart in four steps. This flowchart is described below.
Flowchart Description:
The flowchart is generated running the Binary_features_generation.ipynb file, and can be viewed directly in the repository. It graphically represents the four main steps of this experimental process.
STEP A: Feature Generation
The name of the Jupyter Notebook is Binary_features_generation.ipynb.
It presents the following subsections: image visualization, feature extraction, and binary concatenation of files in csv format. Folders /IMAGES/ and /FEATURE_RESULTS/ are referenced, and the feature_*.csv files are introduced.
STEP B: Optimization of QVM Model Parameters
The name of the Jupyter Notebook is FIT_DP_NODP_CIRCUIT.ipynb.
It presents the following subsections: library import, function definition, data loading, normalization, quantum circuit definition, cost function, COBYLA optimization, and model evaluation. Reference is made to Table 1 and to the optimal parameters stored in the variable named opt_var.
STEP C: Verification on Quantum Hardware/Simulation
The name of the Jupyter Notebook is QVM_verification_two_qubits.ipynb.
It presents the following subsections: environment setup, backend selection, data and parameter loading, circuit definition, transpilation, circuit execution, and results analysis. Reference is made to Table 3 and the results comparison graphs.
STEP D: Execution of QKM Model
The name of the Jupyter Notebook is QKM_verification_two_qubits.ipynb.
It presents the following subsections: execution of the QKM model on real quantum computers, real quantum computer execution, quantum kernel estimation algorithm and SVM with QKM for two, three and eight qubits.”
Figure 18. Overall flowchart and Jupyter Notebooks.
- Practical applicability of the method
REVIEWER: The author implemented fault diagnosis based on optical partial discharge spectra, but in practice, how can the discharge spectra of transformer oil gaps be obtained? Therefore, this method is difficult to apply to on-site transformer fault diagnosis.
ANSWER:
Thank you for your appreciation.
Indeed, in-situ experimental imaging of transformer oil spaces is extremely complex. In this work, the tests have been performed in the laboratory with oil samples extracted from the transformer.
Consider that the main objective of this work is to explore the feasibility and potential of quantum machine learning models for the classification of electrical discharges in dielectric oils, using a controlled laboratory environment.
Although our current work is not focused on in-situ implementation, it could lay the groundwork for future research in that direction. This laboratory set-up offers the advantage of immunity to electromagnetic interference.
Conclusion
We hope that these answers clarify your comments. Besides, we are convinced that the publication in a Zenodo repository where the images, Jupyter Notebooks, and a detailed README are included, as well as the modifications made to the manuscript, will significantly improve the quality of the article.
We thank the reviewer again for his valuable comments.

Reviewer 2 Report
Comments and Suggestions for Authors
(1)It is not clear whether the overall focus of the research work is on methods or on the feasibility of practical applications. There are too many theoretical quantum machine learning methods involved, and they are not the author's original innovation.
(2)The abstract section mentions a comparison of accuracy, but there is no clear description of the composition and division of the fault type classification and modeling dataset.
(3)The manuscript is divided into too many paragraphs, which is not conducive to readers' understanding. It is suggested to add an overall flowchart of the data.
(4)All the figures are relatively vague and the description in the paper is not detailed enough, such as what the functions of each part are in Figure 1 (b); How are Figures 12, 15, 16, and 17 constructed.
(5)There are many writing irregularities, such as the retention of significant figures in tables.
Author Response
REVIEWER 2
Thank you very much for your time and valuable feedback on our work "Quantum Variational vs Quantum Kernel Machine Learning Models for Partial Discharge Classification in Dielectric Oils". We have carefully reviewed your suggestions and responded to each of the proposed points. Furthermore, for clarity and reproducibility, we have created a public repository on Zenodo containing all the data we have generated for this paper, the source code developed in Python, and a detailed README explaining step by step the experimentation process.
Currently private link to the Zenodo repository:
[29]
https://zenodo.org/records/14589724?preview=1&token=eyJhbGciOiJIUzUxMiJ9.eyJpZCI6IjcxYjY0NmJkLTVhN2MtNGExZC1iMTM1LTRkYzJkZDBhODlkNCIsImRhdGEiOnt9LCJyYW5kb20iOiIwNTg5ZmIzODg4ODMzODY3N2IxYmZiNGRmZGJkODJlNCJ9.Cuy1bJrWskY0Z0ZeQkJcVyx05hipFSLE797lYY-y7hI2GTBir1dIXhlDX3zIeEYYaVO2pFaE8nVYT2poUZnF8w
Reference to the final public repository in Zenodo:
[29] Monzón-Verona, J. M., García-Alonso, S., & Santana-Martín, F. J. 2024. Software and dataset of Quantum Variational vs Quantum Kernel Machine Learning Models for Partial Discharge Classification in Dielectric Oils. Zenodo. https://doi.org/10.5281/zenodo.14589724
(1) Methods, practical applications, innovation.
REVIEWER: It is not clear whether the overall focus of the research work is on methods or on the feasibility of practical applications. There are too many theoretical quantum machine learning methods involved, and they are not the author's original innovation.
ANSWER:
We appreciate the reviewer's comment.
We have revised and modified both the abstract and the introduction to more clearly emphasize the contribution and novelty of our work.
- Main contribution: Our work demonstrates, for the first time, the application and comparison of two quantum machine learning models (QVM and QKM) for the classification of partial discharge (PD) images in dielectric oils, using real data obtained with a high-resolution optical sensor. Furthermore, the impact of increasing the number of qubits on the accuracy of the QKM model is explored.
• Novelty:
- Application to a real problem: unlike previous studies that focus on simulated or theoretical data sets, our work is applied to a real and practical problem in the field of dielectric oil diagnostics of transformers, using real PD images.
- Comparison of QVM and QKM: a comprehensive comparison is made between the QVM and QKM models in the specific context of PD classification, showing the advantages and limitations of each.
- Study of the impact of the number of qubits in QKM: It is shown that increasing the number of qubits in the QKM model significantly improves the classification accuracy of the four binary combinations of PD classes.
- Using real quantum computers: the models were implemented and run on three IBM fault-tolerant quantum computers, demonstrating their operation on real quantum hardware and providing results comparable to classical simulations.
- Transparency and reproducibility: a repository has been created on Zenodo, with a detailed README, where the images of the electric discharges used, the Jupyter Notebooks for the extraction of the features, and the Jupyter Notebooks of the QVM and QKM have been published, all in order to facilitate the transparency and reproducibility of our results, so that the scientific community can access and use them.
In-situ experimental images of transformer oil spaces are extremely complex. In this work, the tests have been performed in the laboratory with oil samples extracted from the transformer.
Consider that the main objective of this work is to explore the feasibility and potential of quantum machine learning models for the classification of electrical discharges in dielectric oils, using a controlled laboratory environment.
Although our current work is not focused on in-situ studies, it lays the groundwork for future research in that direction. This laboratory set-up offers the advantage of immunity to electromagnetic interference.
Therefore, on line 43 we have introduced the following text:
“In-situ experimental images of transformer oil spaces are extremely complex. In this work, the tests have been performed in the laboratory with oil samples extracted from the transformer. Consider that the main objective of this work is to explore the feasibility and potential of quantum machine learning models for the classification of electrical discharges in dielectric oils, using a controlled laboratory environment.
Although our current work is not focused on in-situ studies, it could lay the groundwork for future research in that direction. This laboratory set-up offers the advantage of immunity to electromagnetic interference.”
In addition, on line 183 we have introduced the following text:
“The main contribution of this paper is that for the first time two quantum machine learning models, QVM and QKM, are applied and compared for the classification of electrical discharge images in dielectric oils, using real data obtained with a high-resolution optical sensor.
The novelty of this work can be summarized in the following points. The work is applied to a real problem, unlike other previous studies that focus on simulated or theoretical data sets. In addition, a study is carried out on the impact of the number of qubits in QKM and it is shown that increasing the number of qubits in this model significantly improves the accuracy in the classification of the four binary combinations of the classes. On the other hand, real quantum computers are used and the models are implemented and executed on three fault-tolerant IBM quantum computers, demonstrating their operation on real quantum hardware and providing results comparable to classical simulations. This work also provides transparency and reproducibility by creating a repository on Zenodo [29], with a detailed README, where the images of the electrical discharges used, the Jupyter Notebooks for the extraction of the features, and the Jupiter Notebooks with the Python programming of QVM and QKM with the respective figures have been published, so that the scientific community can access and use them.”
(2) Abstract and data set
REVIEWER: The abstract section mentions a comparison of accuracy, but there is no clear description of the composition and division of the fault type classification and modeling dataset.
ANSWER:
We appreciate the reviewer's comment.
Taking into account the clarifications introduced in the reviewed version, the abstract has been modified as follows:
“Abstract: In this paper, electrical discharge images are classified using AI with quantum machine learning techniques. These discharges are originated in dielectric mineral oils and were detected by a high-resolution optical sensor. Two quantum binary classification models were developed and compared for four discharge binary combinations. The first was a quantum variational model (QVM), and the second a conventional support vector machine (SVM) with a quantum kernel model (QKM). The execution of these two models was realized on three fault-tolerant physical quantum IBM computers. The novelty of this article lies in its application to a real problem, unlike other studies that focus on simulated or theoretical data sets. In addition, a study is carried out on the impact of the number of qubits in QKM and it is shown that increasing the number of qubits in this model significantly improves the accuracy in the classification of the four binary combinations studied. In the QVM, with two qubits, an accuracy of 92% was observed in the first discharge combination in the three quantum computers used, with a margin of error of 1% compared to the simulation obtained on classical computers.”
To clarify the description of the composition and division of the fault type classification and modeling data set we have created a public repository on Zenodo [29] containing all the data set we have generated for this paper, the source code developed in Python, and a detailed README explaining step by step the experimentation process.
The images used for the four classes ARC, BREAK, NO_PD and PD, which appear in the repository, come from a previous work [30] carried out by our research team. These images are the basis for generating new data from which features are extracted using a Python program within the file "binary_features_generation.ipynb" contained in the repository.
(3) Flowchart
REVIEWER: The manuscript is divided into too many paragraphs, which is not conducive to readers' understanding. It is suggested to add an overall flowchart of the data.
ANSWER:
We appreciate your suggestion.
A new section 6 has been added to explain the overall flowchart that explains the main steps followed in the article to obtain the different results shown in sections 2 to 5.
The following explanation has been added to the article on line 723:
“6. Overall flowchart
In this section an overall flowchart, see Figure 18, that explains the main steps followed in the article to obtain the different results shown in sections 2 to 5 has been added. We believe this will make the procedure easier to understand.
A detailed description of each step of the process is included in the README of the Zenodo repository [29], along with links to the corresponding Jupyter Notebooks. To make it easier to understand the overall workflow, we have included a summary description of the flowchart in four steps. This flowchart is described below.
Flowchart Description:
The flowchart is generated running the Binary_features_generation.ipynb file, and can be viewed directly in the repository. It graphically represents the four main steps of this experimental process.
STEP A: Feature Generation
The name of the Jupyter Notebook is Binary_features_generation.ipynb.
It presents the following subsections: image visualization, feature extraction, and binary concatenation of files in csv format. Folders /IMAGES/ and /FEATURE_RESULTS/ are referenced, and the feature_*.csv files are introduced.
STEP B: Optimization of QVM Model Parameters
The name of the Jupyter Notebook is FIT_DP_NODP_CIRCUIT.ipynb.
It presents the following subsections: library import, function definition, data loading, normalization, quantum circuit definition, cost function, COBYLA optimization, and model evaluation. Reference is made to Table 1 and to the optimal parameters stored in the variable named opt_var.
STEP C: Verification on Quantum Hardware/Simulation
The name of the Jupyter Notebook is QVM_verification_two_qubits.ipynb.
It presents the following subsections: environment setup, backend selection, data and parameter loading, circuit definition, transpilation, circuit execution, and results analysis. Reference is made to Table 3 and the results comparison graphs.
STEP D: Execution of QKM Model
The name of the Jupyter Notebook is QKM_verification_two_qubits.ipynb.
It presents the following subsections: execution of the QKM model on real quantum computers, real quantum computer execution, quantum kernel estimation algorithm and SVM with QKM for two, three and eight qubits.”
Figure 18. Overall flowchart and Jupyter Notebooks.
(4) Figures and descriptions
REVIEWER: All the figures are relatively vague and the description in the paper is not detailed enough, such as what the functions of each part are in Figure 1 (b); How are Figures 12, 15, 16, and 17 constructed.
ANSWER:
We appreciate the reviewer's comment.
We have revised the figures and their descriptions in the text to clarify them.
- In Figure 1b: The experimental setup shown in Figure 1b was developed in the recently published paper by the authors with reference [31]. This reference is explicitly included on line 210. Previously it only appeared at the bottom of Figure 1b and this was not sufficient.
- In Figures 12, 15, 16 and 17: A more detailed explanation of how these Figures were obtained has been added. In addition, it has been specified that these figures represent kernel matrix, and their generation process has been described, including the link to the Zenodo repository.
The following texts have thus been included:
On line 603:
“The exact kernel results for the PD_BREAK class combination are depicted in Figure 12a. Only the upper diagonal needs to be computed, since the matrix is symmetric. The element kernel matrix obtained is 580x580 as can be seen in the Data Index (X1) and (X2). The number of computations, which corresponds to the quantum circuit in Figure 11, is determined by Equation (26). This circuit was used to estimate the kernel on a quantum computer, performing 168,200 evaluations. For each evaluation the kernel matrix value is normalized between 0 and 1.
Figure 12b shows a comparative study between the exact kernel value and the value obtained with IBM Kyoto computer for row 40 and columns 0 to 24 (Pub 0 to Pub 24) of the matrix shown in 12a. The matrix was generated using the library Qiskit [28], with the Jupyter Notebook QKM_verification_two_qubits.ipynb allocated in the repository [29].”
On line 648:
“The exact kernel results for the PD_NOPD, BREAK_NOPD and PD_ARC class combinations are depicted in Figure 15a 16a and 17a, respectively. Only the upper diagonal needs to be computed, since the matrix is symmetric. The element kernel matrix obtained is 580x580 as can be seen in the Data Index (X1) and (X2). The number of computations, which corresponds to the quantum circuit in Figure 11, is determined by Equation (26). This circuit was used to estimate the kernel on a quantum computer, performing 168,200 evaluations. For each evaluation the kernel matrix value is normalized between 0 and 1.
Figures 15b 16b and 17b show a comparative study between the exact kernel value and the value obtained with IBM Osaka, Brisbane, IBM Kyoto computers, respectively, for row 40 and columns 0 to 24 (Pub 0 to Pub 24) of the matrix shown in Figures 15a, 16a and 17a, respectively. These matrices were generated using the library Qiskit [28], with the Jupyter Notebook QKM_verification_two_qubits.ipynb allocated in the repository [29].”
(5) Writing irregularities
REVIEWER: There are many writing irregularities, such as the retention of significant figures in tables.
ANSWER:
We appreciate the reviewer's comment.
We have carefully revised the manuscript and corrected some irregularities in the writing.
On line 677 where it says “between 2.58 s and 2.63 s” it should say “between 2,587 s and 2,639 s”.
On line 708 where it says “between 2.352 s and 4.229 s” it should say “between 2,352 s and 4,229 s”.
In Tables 4, 5 and 6 the separator “,” has been introduced in some figures that have more than three significant figures.
The presentation of the results in the tables has been revised to ensure that an adequate number of significant figures are used. In particular, the numerical precision in Table 1 has been reduced to two decimal places because new simulations have been performed and with those two decimal places the accuracy is maintained.
Conclusion:
We hope that these answers clarify your comments. Besides, we are convinced that the publication in a Zenodo repository where the images, Jupyter Notebooks, and a detailed README are included, as well as the modifications made to the manuscript, will significantly improve the quality of the article.
We thank the reviewer again for his valuable comments.

Round 2
Reviewer 1 Report
Comments and Suggestions for Authors
Thanks for adopting my suggestions, I have no further comment.